# Flow Raman Spectroscopy for the Detection and Identification of Small Microplastics

**DOI:** 10.3390/s25051390

**Published:** 2025-02-25

**Authors:** Alexander Kissel, André Nogowski, Alwin Kienle, Florian Foschum

**Affiliations:** 1Institut für Lasertechnologien in der Medizin und Meßtechnik an der Universität Ulm (ILM), Helmholtzstr. 12, 89081 Ulm, Germany; alwin.kienle@ilm-ulm.de (A.K.);; 2Faculty of Natural Sciences, Ulm University, 89069 Ulm, Germany; 3SKZ-KFE gGmbH, Europäisches Zentrum für Dispersionstechnologien (EZD), Weissenbacher Str. 86, 95100 Selb, Germany

**Keywords:** microplastic, Raman spectroscopy, water, particle analysis

## Abstract

The most commonly used methods for the detection and identification of small microplastics generally require a complex sample preparation procedure and only allow for static measurements. Quality control of food and drinking water therefore requires a lot of effort. Especially in view of the increasing amount of plastic waste in the environment, the rising public awareness of the issue and the indications for adverse effects of microplastics on human health, more sophisticated measuring methods are required. In this paper, we present a measuring setup for the detection and identification of microplastics using flow Raman spectroscopy. We demonstrate the ability to acquire Raman spectra of individual particles as small as about 4 µm, enabling the identification of their plastic type. We show measurements of differently generated and shaped particles and particles made of different plastic types, highlighting the observed challenges and differences. Finally, we show possible applications of the measuring method. We demonstrate that the measuring principle is suitable for detecting and identifying microplastic particles among other particles and that aged plastics can still be distinguished by their Raman spectra. Overall, our results show that flow Raman spectroscopy is a promising method that could significantly reduce the effort required to detect microplastics.

## 1. Introduction

It is well known that microplastics have contaminated the environment [1,2,3] as well as food and drinking water resources all over the world [4,5]. Microplastic particles are inhaled and ingested with the food and have been found in the human bloodstream [6], in breast milk [7], as well as in various organs [8]. Recent studies suggest that microplastics may trigger a variety of diseases [8], although the impact of these particles on human health is largely unknown and more research is needed. Especially, small particles with a size of less then 150 µm seem to be relevant for potential health effects, as they can be absorbed by the human body [5,9]. More precisely, particles with a size of less than 20 µm are absorbable by organs, while particles with a size of less than 10 µm can cross the blood–brain barrier and cell membranes [9]. However, due to their small size and chemical inertness, detecting and identifying microplastic particles currently requires significant effort. The preferred and most used identification methods are Fourier transform infrared spectroscopy (FTIR) and Raman spectroscopy or variants thereof, such as stimulated Raman spectroscopy (SRS), coherent anti-Stokes Raman spectroscopy (CARS) and surface-enhanced Raman spectroscopy (SERS) [10,11,12]. These methods are non-destructive and can enable measurements of particles significantly smaller than 1 µm [13,14]. When analyzing a water sample, the sample is usually filtered first and then the filter is statically analyzed [12]. This procedure is not only time-consuming, but also prone to contamination [12]. It has been shown that airborne particles can contaminate the samples; a clean room environment is therefore required for examinations [12]. The currently used measuring procedures also do not allow for real-time, autonomous or continuous measurements, limiting their applications [15]. Microplastic research could benefit substantially from new measuring methods with less laborious measuring procedures and lower costs, especially from devices capable to detect and identify small particles with a size of less than 150 µm.

We see the development of flow through measuring systems as key in this context. With these, the time-consuming sample preparation process could potentially be avoided. By measuring the particles directly in the liquid to be tested, the contamination-prone and time-consuming process of sample filtering will become unnecessary. Flow through measuring setups could also allow for real time and continuous measuring applications. Regarding CARS and SRS setups, detection of particles in the flow smaller than ≈40 µm [16], ≈10 µm [17] and 1 µm [18] has been presented in previous work. With much less complex and expensive spontaneous Raman spectroscopy setups, the detection of ≈100 µm-sized particles [2] and ≈26 µm [19] in the flow has been successfully demonstrated. In this work, we present a spontaneous Raman spectroscopy measuring setup, capable of detecting and identifying ≈4 µm-sized microplastics in the flow. We perform measurements with research particles as well as non-spherical plastic fragments. We present the differences and difficulties when measuring these particles. Lastly, we demonstrate that we can identify microplastic particles among the vast majority of other particles in river water samples.

## 2. Materials and Method

### 2.1. Samples and Sample Preparation

The developed measuring system was tested and validated with various microplastic particles. These particles mainly differed in their plastic type, their size distribution and their shape. Research particles were used to evaluate the detection limit and the general behavior of the measuring system. Differently generated plastic particles were produced to test the measuring system in a more application-oriented manner. On the one hand, spherical particles were produced by melt dispersion, and on the other hand, abraded particles were produced by dry and wet abrasion. In order to have a reference for the flow measurements, the Raman spectra of the examined plastics (new and aged) were also measured on a commercially available Raman microscope (alpha300 R, WITec GmbH, Ulm, Germany). Particles made of polyethylene (PE), polystyrene (PS), polypropylene (PP), polyethylene terephthalate (PET), polylactic acid (PLA) and polymethyl methacrylate (PMMA) were used in the investigations. The Raman spectra of the plastics, the microplastic particles themselves, and how the particles were generated are presented in the following.

Research particles made of PS and PMMA (microparticles GmbH, Berlin, Germany) were used to evaluate the behavior of the measuring system with differently sized particles and to determine the lower detection limit. While the spherical PS particles were 3.97 ± 0.06 µm, 15.59 ± 0.13 µm, 19.3 ± 0.33 µm and 50.7 ± 0.7 µm in size, the spherical PMMA particles had a diameter of 10.39 ± 0.26 µm and 101.0 ± 1.8 µm. Among the tested particles, the research particles were the only ones that were perfectly spherical and with such a small size distribution. This allowed for an evaluation of the detection limit. On the left-hand side of Figure 1, PS research particles with a diameter of 19.3 µm are exemplarily shown.

Dry-abraded particles were obtained by a sanding process. Injection-molded plastic plates were sanded and the particles collected. The resulting non-spherical particles were dispersed in water with a surfactant and finally filtered at 112 µm. For wet abrasion, a new method was developed and trialled within the project for the defined stressing of plastic workpieces. The plastics were exposed to ultrasound-induced cavitation in an ultrasonic bath (Elmasonic P 60H, Elma-Hans Schmidbauer GmbH, Germany). The unaged and aged plastic plates (40 × 40 × 4 mm) were sonicated in particle-free water (Ampuwa^®^, Fresenius Kabi, Bad Homburg, Germany) in a cleaned screw-top bottle. This prevented contamination of the sample chamber and the test specimens. Defined and reproducible sonication for 30 min was ensured by a fixed position in the ultrasonic bath. Particles made of PP, PET, PLA, PE and PS were used for the measurements. In Figure 1, some abraded particles are exemplarily shown on the second and third picture from the left. A single abraded PP particle and an abraded particle made of PET are displayed. It can be seen that the shape of the particles differed from the research particles and also differed between the plastic types. The abraded particles featured a wide variation in their volume. Most of the particles were significantly smaller than the pore sizes used to filter the samples, although some particles also exceeded the pore sizes in one dimension. Nevertheless, the volume of the abraded particles was generally less than that of similarly sized research particles. The abraded particles tended to appear in the form of thin flakes and therefore had a rather one-dimensional shape.

Lastly, the measuring system was also tested with spherical PP microparticles. These were produced by a melt dispersion technique on a twin-screw extruder (MiniLab Mikro-Compounder, Thermo Scientific, Waltham, MA, USA). For this purpose, the plastics were mixed with the water-soluble component polyethylene glycol (PEG) and compounded so that the plastic was present in spherical form as a disperse phase in the PEG melt. The surrounding water-soluble matrix in the cooled compound was afterwards dissolved in water. A PEG-free liquid matrix was achieved by repeatedly washing out and filtering the plastic particles. In Figure 1, it can be observed that these particles resembled the research particles, though they were less spherical. Their size distribution also was much larger.

To determine the reference Raman spectra of the investigated plastics, measurements with an alpha300 R WITec Raman microscope were performed. The Raman spectra were recorded on solid non-colored plastic sheets made of the different plastic types. As microplastic in the environment is exposed to environmental influences such as UV light, and to allow for a more application-oriented evaluation of the presented measuring method, two samples were measured for each plastic type, one of which was artificially aged. For this purpose, plates of five unfilled plastics (PS, PE, PP, PET and PLA) were produced by injection-molding. After production, these plates were aged in a weathering instrument (Xenotest 440, Atlas Material Testing Technology GmbH, Germany) in accordance with EN ISO 4892-2:2013+A1:2021 [20]. The plastic plates were exposed to a process of irradiation with xenon-arc lamps as the light source for 1000 h. The detailed conditions of temperature, humidity and wetting are described in the Standard [20]. Possible differences in the Raman spectra due to plastic aging were examined in this way. The recorded Raman spectra are displayed in Figure 2. For each plastic type, the spectra of an aged and an unaged sample can be seen. All plastics had distinctive Raman bands, allowing for precise differentiation of the plastic type. Apart from a slight increase in fluorescence with PET, no significant changes due to aging were observed.

### 2.2. Measurement Setup

The measuring principle of the developed setup is schematically shown in Figure 3. The light path to excite the Raman scattering starts with the 532 nm laser (RLTMLL-532-5W-5, Roithner Lasertechnik GmbH, Vienna, Austria), which had a measured nominal power of 5.66 W. The laser wavelength was chosen as a trade-off to reduce fluorescence and at the same time to obtain a strong Raman signal. We have not observed any problems caused by fluorescence on the examined particles so far. The selected laser wavelength was well suited for the non-colored particles. We note, however, that a wavelength of 785 nm may present problems with fluorescent particles. Two deflection mirrors and a laser-mounted variable polarization attenuator were used to adjust the laser beam. As these parts are not relevant for the measuring principle, they are not displayed in Figure 3. After the deflection mirrors, the beam propagated through a 550 nm cut-off dichroic short-pass beam splitter (DMSP550R, Thorlabs, Newton, NJ, USA) and was focused into the 2 mm × 2 mm flow cell (Type 77, FireflySci Inc., Northport, NY, USA) by an f = 20 mm lens. A beam trap absorbed the laser after the cuvette to reduce stray light. The specific laser parameters determined by the manufacturer prior to delivery allow for calculation of the theoretical focus beam waist in the flow cell. With a (1e2) laser beam width d = 2.512 mm × 3.375 mm, a beam propagation ratio of $M^{2}$ = 3.87, a wavelength of the laser λ = 532.2 nm and a focus length *f* = 20 mm, the minimal beam waist diameter was$$\omega_{0}=\frac{1.27M^{2}\lambda f}{d}=21\;µm.$$

The same lens that focused the laser was also used in the backward direction to image the inelastically scattered light onto the 1 mm fiber of the Raman spectrometer (QE Pro Raman532+, Ocean Optics, Largo, FL, USA). The light was collimated by this lens, reflected by the dichroic mirror and focused on the fiber by a second f = 50 mm lens. An additional 550 nm long-pass filter (FELH0550, Thorlabs) was used to block the excitation laser. The whole optomechanical setup was built on a 600 mm × 800 mm baseplate. Compared to the optical layout of previous flow-through Raman configurations [19,21], our new design has several advantages. By focusing the excitation laser, the beam density was increased, resulting in an enhanced Raman signal. By using the same lens to focus the laser and to image the inelestically scattered light in the backward direction, the focal plane of the laser was also automatically the focal plane of the Raman detection. The required effort for adjustments was reduced significantly. Theoretical investigations further indicate that the Raman emission of spherical particles is dependent on the scattering angle [22]. Detection in the backward direction can thereby offer advantages [22].

Another key feature of our measurement setup is the implementation of an acoustic particle-focusing technique [23,24]. As also displayed in Figure 3, piezoelectric chips (PA3BCW, Thorlabs) were attached to the outside walls of the 2 mm × 2 mm flow cell. Driven with a function generator (MSO5104, Rigol Technologies Inc., Suzhou, China) and a power amplifier (HVA200, Thorlabs), the piezos were used to generate a standing acoustic wave inside the canal of the cuvette. As schematically shown in Figure 4, the acoustic force moves particles toward the node of the waves. Depending on the amount of nodes, single or multiple focus points are possible [24]. The required frequency *f* for a single node *N* = 1, and therefore a single focus point in the middle of the canal, can be calculated with$$f=\frac{c_{m}N}{2d}=371\;\mathrm{kHz},$$
whereby $c_{m}$ = 1484 msec represents the speed of sound in water and *d* = 2 mm, the canal width [24].

In order to focus the particles in the center of the flow-through channel, the piezoelectric chips were applied on two orthogonal outer walls of the cuvette. In total, four piezoelectric chips were used, two for each side. Superglue was used to attach the chips. The piezos were all wired in series and driven with a sinusoidal voltage of 0–100 V. As the speed of sound in water varies with temperature and the width of the flow-through channel slightly deviated, the optimal frequency for focusing was found to be 380 kHz. Figure 5 shows the flow cell in its mount (left), as well as video snapshots of focused and unfocused research particles in flow (right). In this case, the piezos were mounted only on one side of the cuvette and the particles were focused in one dimension. In order to achieve better visibility of the particles, differential images were generated to eliminate all static impurities and reflections. The images are furthermore contrast-enhanced for the same reason. It can be seen that the piezos were only attached in the lower section of the cuvette; the particles were, therefore, only actively focused in this section. However, it can be seen that the particles maintained their position in the center of the channel as they continued to flow through the cuvette.

Focusing the particles had several advantages. We consider the focusing of the Raman excitation laser and a high numerical aperture of the Raman detection unit as the key to high sensitivity and to achieve Raman measurements of small particles. However, this drastically reduces the possible depth of the field and results in a small measuring volume. Using a smaller flow-through channel to match the measuring volume results in a higher flow rate, which limits the possible integration time and therefore also reduces the sensitivity. When using a larger channel, to allow for longer integration times, the probability of a particle passing through the measuring volume is extremely low without particle focusing. Therefore, a much larger sample volume is required. With acoustic focusing, we can achieve a high sensitivity and can at the same time measure a large proportion of the particles. The same conflict of objectives is also known in the field of fluorescent cell cytometry and SRS, where particle-focusing methods have also been adapted [17,25]. The acoustic focusing also helped with another observed problem. Particles frequently adhered to the channel walls of the flow cell, resulting in intense scattering near the measurement location. The acoustic focusing prevented particles from adhering to the channel walls.

A peristaltic pump (iPump 2F, Landgraf Laborsysteme HLL GmbH, Langenhagen, Germany) was used to pump the samples through the cuvette. Based on the results of Kniggendorf et al. [21], PTFE tubes were used to minimize particle adhesion on the inner surfaces of the tubes. The setup was rinsed with purified water (Ampuwa^®^, Fresenius Kabi, Bad Homburg, Germany) before and after each measurement. The flow direction was chosen so that the samples were sucked in and passed through the cuvette to be measured first. The samples then flowed through the pump, after which they were disposed of.

## 3. Results

In the following, measurements with the developed flow-through Raman spectrometer are presented. This includes measurements of microplastic suspensions in purified and treated water (Section 3.1), as well as measurements with other suspended solids in the water and soiled river water samples (Section 3.2). The research particles, abraded particles and melt-dispersed particles were used to characterize the measuring system. These defined particles, and the absence of contaminants in the water, allowed for an evaluation of the detection limit of the measuring system, as well as study of different types of plastics and differently shaped particles. Subsequently, the robustness of the measuring principle and the ability to detect a microplastic particle among other particles was demonstrated with soiled water samples.

As the particles were suspended in water during our measurements, the Raman spectra of the water, shown in Figure 6, were recorded as a background in each measurement. When compared with Figure 2, it can be seen that the Raman bands partially overlap with those of the investigated plastics. For this reason, the background of the water and possible other residues was subtracted before each measurement series by taking a dark image.

### 3.1. Measurements of Pure Microplastic Suspensions

To measure the particles, the procedure of Glöckler et al. [19] was applied. A suspension of the microplastic particles in purified water (Ampuwa^®^, Fresenius Kabi, Bad Homburg, Germany) was prepared and pumped through the system. The flow rate was set to 0.5 mLmin for all measurements, resulting in a particle velocity of approximately 2.1 mms in the flow-through channel. Spectra were recorded over a period of 60 s; a volume of 0.5 mL therefore flowed through the cuvette during each measurement. The integration time of the Raman spectrometer was set to 10 ms for each spectrum. As the particle concentration was not defined, it varied between the samples. In Figure 7, exemplary measurements of PS research particles (top) and abraded PP particles (bottom) are shown. For both measurements, the spectra of an individual particle acquired in the flow (left) and the measured intensity at three Raman bands in the time domain are shown (right). No Raman bands of eventual impurities were observed when purified water was used for the suspensions and only the expected spectra of the added plastics were recorded. In the single spectra, prominent Raman bands at 1012 ${\mathrm{cm}}^{-1}$, 1612 ${\mathrm{cm}}^{-1}$ and 3063 ${\mathrm{cm}}^{-1}$ are visible for PS; Raman bands at 2850 ${\mathrm{cm}}^{-1}$, 2890 ${\mathrm{cm}}^{-1}$ and 2961 ${\mathrm{cm}}^{-1}$ can be seen in the PP spectra. In the time domain, intensity peaks are visible at the expected Raman bands. These are caused by individual particles flowing through the measuring volume one after the other. Raman spectra were detected on all examined particle and plastic types.

As expected, the volume-dependent Raman emission was weaker for smaller particles. It was observed, however, that the intensity, and thus the signal-to-noise ratio (SNR), varied significantly between the samples. Abraded particles showed a weaker signal than melt-dispersed or research particles of similar size. There are two explanations for this. Firstly, the abraded particles varied a lot in their volume, with the majority of the particles being much smaller than the pore sizes used to filter the samples. Secondly, as explained in Section 2.1, the abraded particles generally appeared in the shape of thin flakes. Their volume was much smaller than the spherical research and melt-dispersed particles, even when they exceeded the size of the latter in one dimension.

Also visible in Figure 7 is the varying signal of the spherical research particles with a small size dispersion. This can be explained by the fact that even though we acoustically focused the particles, not all particles passed exactly through the focal point of the detection and excitation lens. Particles that are outside the focal point are recorded with a weaker intensity. In addition, the spectra were integrated over different time intervals of the particle passage through the measuring volume. This may also have led to intensity differences. However, the size of the particles can be measured by elastic light scattering [19].

The signal intensity proved to be highly dependent on the particle volume, the flow rate and the integration time of the spectrometer. With an adapted flow rate of 0.3 mLmin and an integration time of 40 ms, measurements of PS research particles with a diameter of 3.97 µm were possible. This proved to be the detection limit. The recorded spectra of such a particle is shown in Figure 8.

### 3.2. Measurements of Soiled Suspensions

In Figure 9, the Raman spectra of a single PE particle (left) and a single PP particle (right) recorded in the flow are shown. The typical Raman bands of PE, at 2845 cm^−1^ and 2880 cm^−1^, and the typical Raman bands of PP are visible. These particles were not suspended in purified water, but in exemplary water samples with various contaminants. The PE particle was released out of a weathered plastic plate which was made of PE and contained a portion of 10 % CaCO_3_ as a filler. The newly developed method using ultrasonic stress was used to generate the particle. The measured suspension therefore contained PE as well as CaCO_3_ particles. The shown PP particle was added to river water, which was filtered with a 200 µm filter. The resulting suspension therefore contained particles from the river, especially of biological origin and sediments, and the added microplastic particles. The sample was collected in the Selb river (GPS: 50°10′05.8″ N 12°07′35.8″ E). No microplastic particles were detected in the river water when no microplastics were manually added, most likely due to the small sample volume. Measurements in these soiled suspensions showed much more background noise, caused by the other particles in the water, than measurements in purified water. The other particles in the water showed fluorescence that overlapped with the Raman bands of the plastic particles. It can be seen, however, that the Raman bands of the microplastics are still detectable in the background noise. Single spectra of the plastics were selected for display. When compared with Figure 7, the much higher background noise can be seen.

With these measurements, we were able to show that the measuring principle allows the detection of individual relevant particles among a vast majority of other particles. In our case, no sample preparation was necessary, apart from filtering out coarse impurities. As the relevant spectra were determined manually in this work, larger sample volumes were an obstacle. With an evaluation algorithm that automatically detects relevant spectra in the background noise, larger sample volumes and continuous measurements seem possible.

## 4. Outlook and Conclusions

To the best of our knowledge, we have presented for the first time a measurement setup that enables the acquisition of Raman spectra of particles smaller than 10 µm in flow without the use of nonlinear Raman techniques such as SRS or CARS. Furthermore, we were able to carry out measurements at a flow rate of 0.5 mL/min, which represents a significant increase over previous nonlinear Raman spectroscopy setups with comparable sensitivity. As we record complete spectra, unlike SRS, for example, our measurements also contain much more information about the particles. At the same time, the developed measuring system is considerably more cost-efficient and less complex than nonlinear Raman spectroscopy setups. For the first time, we performed flow measurements of river water samples and with particles of different origins.

Our investigations demonstrated that flow-through Raman spectroscopy is a very promising method for the detection of microplastics. We were able to show that the Raman spectra of plastics are not significantly affected by plastic aging, with distinguishable Raman spectra being recorded for all investigated plastic types and particle origins. Observed differences and difficulties when measuring spherical research particles and non-spherical particles generated by abrasion or ultrasonic stress were presented. Although we were able to detect spherical research particles down to 3.97 µm in diameter, we regard further sensitivity improvements as necessary to reliably detect such small environmental particles. By measuring microplastic particles in river water samples, we were able to demonstrate the robustness and potential for application of the developed measuring principle. Apart from filtering for coarse impurities, no prior sample preparation procedures were required for the detection of microplastic particles among a vast majority of other particles.

We recognize many opportunities to further improve the developed measuring system. With implementation of a particle counter and feedback about the particles on which the Raman spectra were acquired, a statistical estimation of the particle composition would be possible. A camera could be used, for example, to observe the entire cross-section of the flow cell to determine the total number of particles as well as their size and shape. The development of an automatic evaluation algorithm could enable continuous and thus larger sample volumes. A longer wavelength of the Raman excitation laser could also offer advantages when measuring fluorescent particles. Any fluorescence is likely to be lower at a longer wavelength, allowing for better measurements of fluorescent particles or particles in a highly fluorescent water background. Also, not all options have yet been fully utilized to achieve maximum sensitivity. By using an oil immersion objective or an aspherical lens as a laser focusing and Raman collimating lens, the numerical aperture could be increased and more of the Raman scattered light could be captured. This would also reduce the beam waist diameter, resulting in more Raman light being excited. As we have not yet observed any thermal effects on the particles, most likely due to the short exposure time in the flow, we also see the possibility of using a more powerful laser for Raman excitation. In this work, the measuring principle was demonstrated with microplastics, however, we also see applications for flow Raman spectroscopy devices in other particle analysis domains, such as cell cytometry. As we were able to detect plastic spheres smaller than the average animal or plant cell, the sensitivity could be sufficient for many applications.

## Figures and Tables

**Figure 1 sensors-25-01390-f001:**
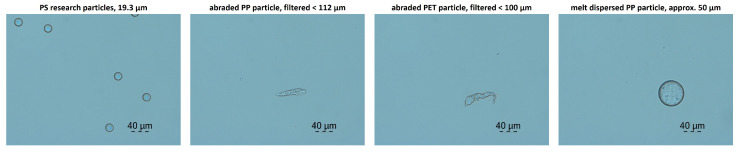
Microscopic images of some of the examined particles. From left to right: spherical PS research particles (19.3 µm diameter), abraded PP particle (filtered to <112 µm), abraded PET particle (filtered to <100 µm) and melt-dispersed PP particle (with a size of approx. 50 µm).

**Figure 2 sensors-25-01390-f002:**
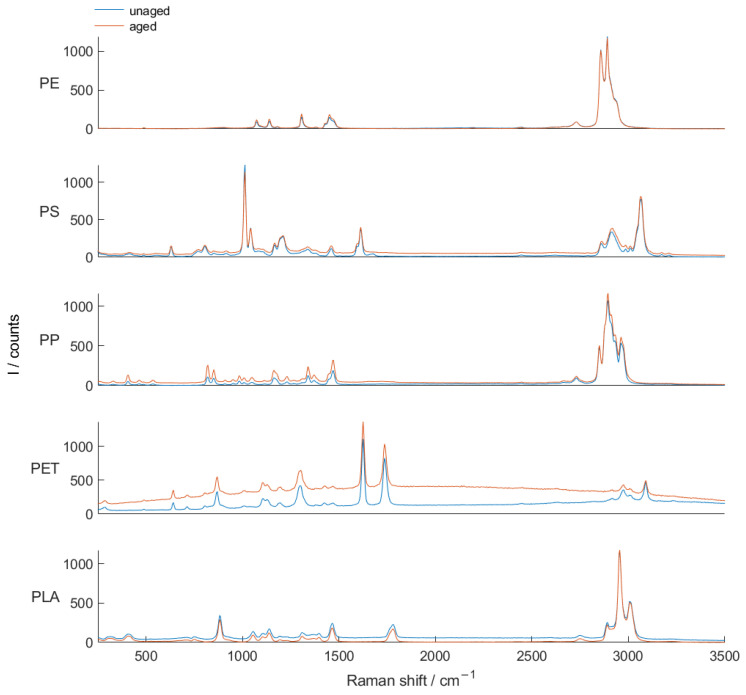
Measured Raman spectra of the investigated aged and unaged plastics.

**Figure 3 sensors-25-01390-f003:**
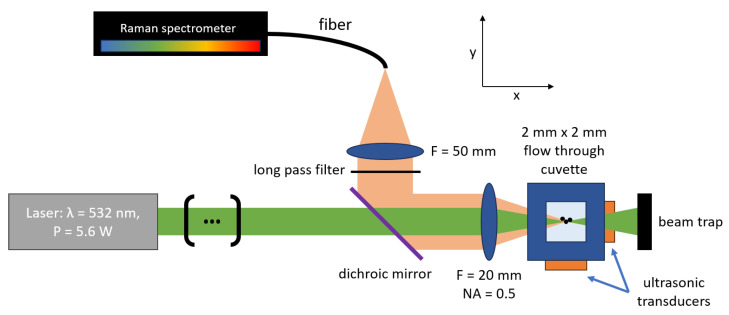
Optical layout of the flow Raman spectroscopy measuring setup. The whole setup was built on a 600 mm × 800 mm baseplate; the flow direction was perpendicular to the visualized plane.

**Figure 4 sensors-25-01390-f004:**
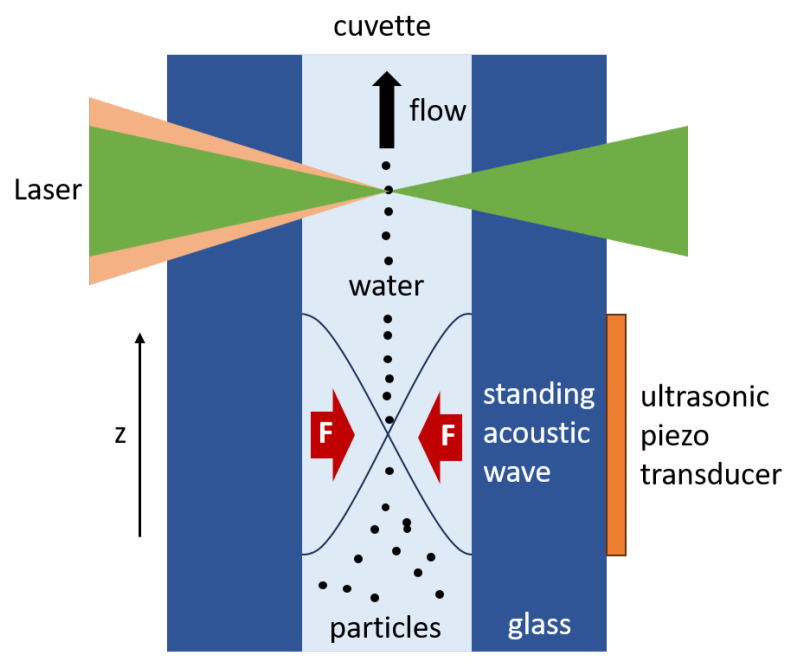
Operating principle of the applied acoustic particle-focusing technique. The piezo transducers were driven with a sinusoidal wave, generated by a function generator. The frequency was adapted to the canal width of the cuvette to produce a standing acoustic wave. The suspended particles in the water are moved to the node of the wave in the center of the cuvette.

**Figure 5 sensors-25-01390-f005:**
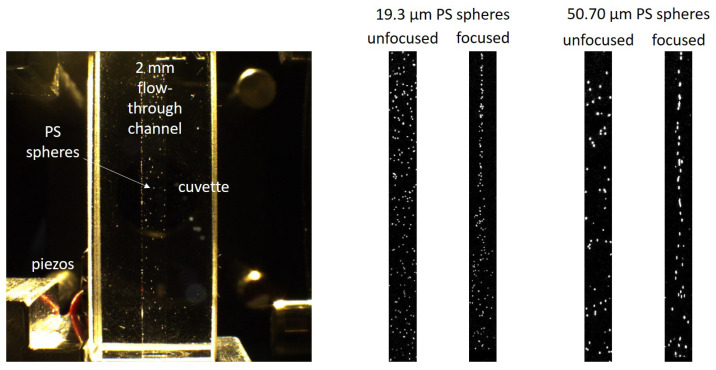
Video snapshots of acoustically focused and unfocused polystyrene spheres with different sizes. A video snapshot of the flow cell in its mount is shown on the left; the difference in the particle distribution with the acoustic focusing switched on and off can be seen on the right side. The images were cropped so that they exactly match the width of the flow channel (2 mm). The flow rate was 0.5 mLmin.

**Figure 6 sensors-25-01390-f006:**
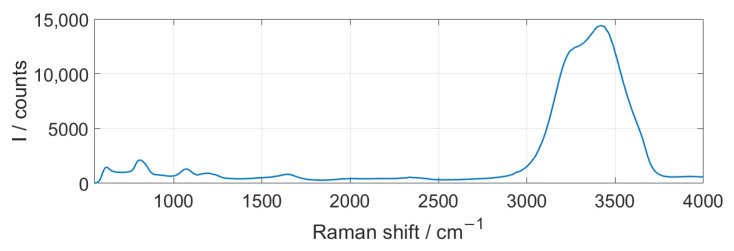
Raman spectra of the water background.

**Figure 7 sensors-25-01390-f007:**
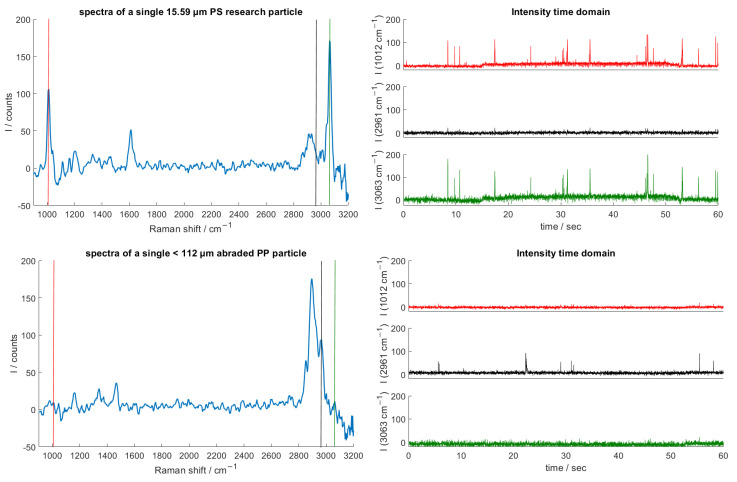
In-flow recorded Raman spectra of a 15.59 µm PS research particle (**top**) and a <112 µm abraded PP particle (**bottom**). In the time domain (**right**), intensity peaks caused by individual particles flowing through the measuring volume can be recognized. The measurements were performed with a throughput of 0.5 mLmin and a spectrometer integration time of 10 ms.

**Figure 8 sensors-25-01390-f008:**
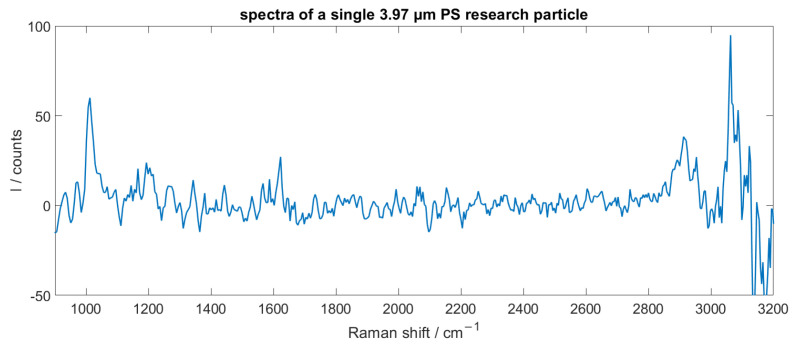
In-flow recorded Raman spectra of a 3.97 µm PS research particle. The measurement was performed with a throughput of 0.3 mLmin and a spectrometer integration time of 40 ms.

**Figure 9 sensors-25-01390-f009:**
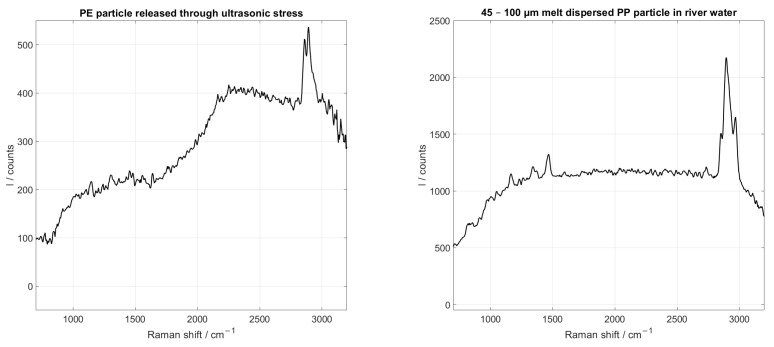
In-flow recorded Raman spectra of a PE particle generated with ultrasonic stress and a PP particle added in river water. The measurements were performed with a throughput of 0.5 mLmin and a spectrometer integration time of 10 ms.

## Data Availability

The original contributions presented in this study are included in the article. Further inquiries can be directed to the corresponding author.

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
