# Peer review of "Flow Raman Spectroscopy for the Detection and Identification of Small Microplastics"

_sensors, 2025, doi:10.3390/s25051390_

Round 1

Reviewer 1 Report

Comments and Suggestions for Authors

A very interesting paper that addresses an important issue - the detection of plastic microparticles in aqueous solutions through the use of Raman spectroscopy. I would like to commend the clear logic and organization of the material presented. The work leaves a positive impression. It is worth noting some comments on the work that do not spoil the overall impression.
1. lines 128-129 how permissible is the laser power of 5.66 W, does it not cause damage to plastic microparticles given the focusing of the laser radiation? What is the permissible limit of power density?
2. line 129 why do you consider 532 nm a compromise option for reducing fluorescence, a number of components have an extremely high fluorescent signal when excited by radiation with this wavelength.
3. In continuation to the previous comment, what should be done when studying colored plastic microparticles? The fluorescent signal when excited by radiation with a wavelength of 532 nm will be several orders of magnitude higher in intensity compared to Ramam scattering. Were preliminary studies carried out with coarse particles?
4. Figure 2, it might be useful to superimpose the graphs on one figure to show how much the spectra overlap.
5. lines 273-274, what exactly does the overlap of Raman bands mean for other plastics? That is, the Raman scattering signal is not visible at all? Why is the fluorescent signal different from water for the two presented particles? How much does the signal in pure water and river water differ for the same particles, it would be useful to provide such a figure in the paper.
Figure 9, the legends on the graphs and the captions to the figure contradict each other, most likely the names of the particles were mixed up.

Author Response

We would like to thank you for your review. Below are our responses to your comments.:

1. lines 128-129 how permissible is the laser power of 5.66 W, does it not cause damage to plastic microparticles given the focusing of the laser radiation? What is the permissible limit of power density?

Answer: 

In our experiments we have not observed any thermal effects on the microplastics. As the power density in the laser focus is indeed very high, this is most likely due to the short exposure time in the flow and the cooling surrounding water. Unfortunately, we cannot estimate the permissible power limit, as it is also most likely highly depended on the plastic type and the absorbance at the laser wavelength.

2. line 129 why do you consider 532 nm a compromise option for reducing fluorescence, a number of components have an extremely high fluorescent signal when excited by radiation with this wavelength.

Answer: 

Thank you for pointing this out. Many fluorescent substances are indeed excited at 532 nm. A high chlorophyll concentration in the water could, for example, make it difficult to detect microplastics in it. An excitation wavelength of 785 nm would offer advantages in this case. The particles we have used for the investigations so far showed no fluorescence. The selected laser wavelength was well suited for these particles. We now note, however, that a wavelength of 785 nm may prevent problems with fluorescent particles. This is now mentioned in the section (line 129-133): “The laser wavelength was chosen as a trade-off to reduce fluorescence and at the same time to obtain a strong Raman signal. We have not observed any problems caused by fluorescence on the examined particles so far. The selected laser wavelength was well suited for the non-colored particles. We note however, that a wavelength of 785 nm may prevent problems with fluorescent particles.”

3. In continuation to the previous comment, what should be done when studying colored plastic microparticles? The fluorescent signal when excited by radiation with a wavelength of 532 nm will be several orders of magnitude higher in intensity compared to Raman scattering. Were preliminary studies carried out with coarse particles?

The investigated particles in this work were all non-colored. As also mentioned above, we did not observe any fluorescence on these particles. We will take a look at colored particles in future work. It is now mentioned in the conclusion, that for fluorescent particles, a longer wavelength for the Raman excitation Laser could provide benefits (line 317-320): “A longer wavelength of the Raman excitation laser could also offer advantages when measuring fluorescent particles. Any fluorescence is likely to be lower at a longer wavelength, allowing for better measurements of fluorescent particles or particles in a highly fluorescent water background.”

4. Figure 2, it might be useful to superimpose the graphs on one figure to show how much the spectra overlap.

Thank you for this recommendation. We tried it, but unfortunately the figure becomes unclear. I don't think it's very relevant either, since all plastics have characteristic bands.

5. lines 273-274, what exactly does the overlap of Raman bands mean for other plastics? That is, the Raman scattering signal is not visible at all? Why is the fluorescent signal different from water for the two presented particles? How much does the signal in pure water and river water differ for the same particles, it would be useful to provide such a figure in the paper.

Figure 9 shows exemplary measurements of microplastic particles, among other particles. The other particles caused a much higher background noise, especially as they were partly fluorescent. The other experiments were carried out in purified water, so the fluorescence of the background is different.  However, it can be seen that the Raman bands of the microplastics are still detectable in the background noise. We see no reason why this shouldn't be the case for other plastics. An automated evaluation algorithm will be necessary to detect the relevant spectra in the background noise and will be part of future work. An evaluation of the sensitivity will also be possible then. Unfortunately, we don't have a measurement of the same particles in pure water on hand.  As the results will be very dependent on the amount of contaminants in the river water and will therefore only apply to the exact sample, we also do not consider the significance to be too high.

6. Figure 9, the legends on the graphs and the captions to the figure contradict each other, most likely the names of the particles were mixed up.

Thank you for pointing this out. The particles were indeed mixed up in the caption of the figure. It is now corrected.

Reviewer 2 Report

Comments and Suggestions for Authors

This works demonstrates a nice strategy to detect and characterize microplastics using flow Raman spectroscopy. The experimental setup is efficient, particularly the acoustic particle focusing technique. The discussion is sound and the main limitations were stressed out, so I reccomend the publication of this manuscript. I have only one minor comment: Figure 6 represents the Raman spectrum of pure water, so, it should has only three bands, at ca. 1630 cm-1 and 3300-3500 cm-1, however, several features were observed below 1300 cm-1. Considering that such spectrum was subtracted as background, I wonder if it influenced the results obtained. I suggest the authors to take a carefull look at this before publication.

Author Response

Thank you for your review. Below we would like to respond to your comment:

Figure 6 represents the Raman spectrum of pure water, so, it should has only three bands, at ca. 1630 cm-1 and 3300-3500 cm-1, however, several features were observed below 1300 cm-1. Considering that such spectrum was subtracted as background, I wonder if it influenced the results obtained. I suggest the authors to take a carefull look at this before publication.

Answer: 

Thank you for pointing this out. Indeed, these raman bands do not appear to originate from the water. Possibly they come from the glass of the cuvette, the lenses, or a possible biofilm in the cuvette or other residues. However, as these were also static, they did not influence our measurements. We now mention this in the manuscript (line 220): "For this reason, the background of the water and possible other residues was subtracted before each measurement series by taking a dark image."